# Combining Optimization and Simulation for Next-Generation Off-Road Vehicle E/E Architectural Design

**DOI:** 10.3390/s24154889

**Published:** 2024-07-27

**Authors:** Cristian Bianchi, Rosario Merlino, Roberto Passerone

**Affiliations:** 1Department of Information Engineering and Computer Science (DISI), University of Trento, 38123 Trento, Italy; 2Iveco Group, 39100 Bolzano, Italy; rosario.merlino@ivecogroup.com

**Keywords:** DSE, off-road vehicles, E/E vehicluar networks, MILP, SIL, AVB, TTEthernet, steer-by-wire, autonomous driving, system-level design

## Abstract

The automotive industry, with particular reference to the off-road sector, is facing several challenges, including the integration of Advanced Driver Assistance Systems (ADASs), the introduction of autonomous driving capabilities, and system-specific requirements that are different from the traditional car market. Current vehicular electrical–electronic (E/E) architectures are unable to support the amount of data for new vehicle functionalities, requiring the transition to zonal architectures, new communication standards, and the adoption of Drive-by-Wire technologies. In this work, we propose an automated methodology for next-generation off-road vehicle E/E architectural design. Starting from the regulatory requirements, we use a MILP-based optimizer to find candidate solutions, a discrete event simulator to validate their feasibility, and an ascent-based gradient method to reformulate the constraints for the optimizer in order to converge to the final architectural solution. We evaluate the results in terms of latency, jitter, and network load, as well as provide a Pareto analysis that includes power consumption, cost, and system weight.

## 1. Introduction

### 1.1. Motivations and Background

The new challenges that vehicle Original Equipment Manufacturers (OEMs) are facing in the design of E/E architectures concern many aspects, ranging from the introduction of Advanced Driver Assistance Systems (ADASs) to the introduction of systems for autonomous driving. In particular, the European General Safety Regulation (GSR) [1] states that, starting from 2024, an off-road vehicle(ORV) must comply with several features in order to be sold on the market. These include, for instance, AIF (Alcohol Interlock Facilitation) [2], BSIS (Blind Spot Information System) [3], CSMS (Cyber Security Management System) [4], DDAW (Driver Drowsiness and Attention Warning) [5], ISA (Intelligent Speed Assist) [6], MOIS (Moving-Off Information Signal) [7], REV (REV Detection) [8], and TPMS (Tyre pressure monitoring heavy duty) [9]. The consequence is the need to integrate high-speed data sensors, new Electronic Control Units (ECUs), and Drive-by-Wire (DBW) technologies [10,11]. Compared to the commercial car market, the *off-road sector* is a difficult niche where the integration of ADAS and the autonomous driving design is challenging for several reasons.

First, ORVs must meet several high-level requirements (RQMTS) such as the high load capacity and the ability to drive in harsh environmental conditions, including steep slopes, rough terrain, and fords, while maintaining high reliability, leading the vehicle to adopt systems not used in conventional cars, including the following:Transfer Case (TC): Transmits power to both the front and rear axles to perform all-wheel drive functionality.Differential-lock systems (DL): Prevents the movement of the front and/or rear differential causing wheel-slip.Power-take-off (PTO): Performs auxiliary functions without an additional engine.Central Tyre Inflation System (CTIS): Optimizes the pressure of the tires given a particular type of terrain.

Other conventional ECUs such as Body Computer (BC), Engine Control Module (ECM), and Electronic Transmission Control (ETC) are also integrated, but with specialized features for the off-road environment.

The second problem is related to the design process, which is often fulfilled using existing technological solutions already adopted and validated in other projects. While initially effective, this approach will become less viable as future vehicle deployments demand increasingly advanced functionalities, leading to worse time to market, performance, cost, and reliability [12]. In this context, efficient communication between different components in a vehicle is crucial. Current vehicular networks based on the interconnected local area network (LIN) and the Controller Area Network (CAN) bus (J1939 standards for heavy-duty vehicles [13]) are unable to support the high data rates required for ADAS and autonomous driving, which means that traditional architectures have reached their load limit. At the same time, it is not industrially acceptable to redesign the entire vehicle infrastructure due to excessive costs and high risk [14].

In this work, we propose a novel design methodology for next-generation electric/electronic (E/E) off-road vehicle (ORV) architectures. Our objective is to move beyond traditional design approaches by considering current regulations and Drive-by-Wire (DBW) technology and by adopting an Ethernet zonal architecture for the new features which require higher data rates. We employ a Design Space Exploration (DSE) framework which leverages pattern-based formal specifications, mixed-integer linear constraints, and optimization to find the best topologies and system implementations [15]. Furthermore, we enhance the framework with domain-specific constraints and functions. Our core contribution consists in the integration of *optimization* and *simulation* techniques. This is beneficial, because during the design process there are real-world deployment aspects that are too complex and hard to model for an optimizer or that simply lead to excessive run times. Conversely, simulation is very effective at handling lower-level details such as protocol handshakes and run-time collisions. Moreover, it allows for us to test the integration of the optimized domain into the entire vehicular architecture, verifying the coexistence of different standards such as CAN and Ethernet. On the other hand, simulation does not scale to a large design space exploration, and only a handful of architectures can be effectively evaluated. The idea we pursue in this work is therefore to combine optimization and simulation. The optimizer selects candidate architectures from a large design space, using relatively abstract models, excluding unfeasible solutions. The simulator therefore examines only a restricted subset, extracting detailed and accurate performance parameters to validate the selection. These parameters are then fed back to the optimizer to refine the solution. The procedure is repeated iteratively to converge to an optimal architecture, thereby maximizing the benefits derived from both techniques. In other words, the optimizer is used to focus the attention to only viable solutions, pruning the design space, while the simulator evaluates the performance for accurate results. As a result, our work greatly improves design automation, where the optimizer generates candidate solutions through different patterns, and simulation results are automatically retrieved.

The paper is structured as follows. Section 1.2 completes and complements the introduction with a comprehensive overview of the state of the art in vehicular networks, Drive-by-Wire technologies, and architectural exploration techniques. Section 2 outlines the methods and the techniques used. In addition, it describes the framework structure, detailing both the optimization and the simulation processes. In Section 3, we evaluate and analyze our framework with an industrial case study, also deriving Pareto fronts and completing a qualitative analysis of the scalability of the methodology to larger case studies driven by autonomous driving data rate requirements. Finally, in Section 4, we draw our conclusions.

### 1.2. State of the Art

#### 1.2.1. In-Vehicle-Networks and E/E Architectures

Since all ECUs in a modern vehicle are interconnected via bus technologies, it is important to understand which solutions are available in terms of data rate, propagation medium, access scheme, data length, supported nodes, topology, and field of application.

The interconnected local area network (LIN) is a low-cost promary–secondary serial protocol with a bus length of up to 40 m. As it is a low-speed protocol, it can be used in low-speed applications [16]. Traditional automotive network architectures rely on the Controller Area Network (CAN) bus to enable communication between ECUs. This bus technology features strong interference immunity, low installation cost, and priority-based collision detection capabilities. However, it is not suitable for the integration of ADAS and autonomous driving capabilities, which require different devices such as cameras, RaDAR, and LiDAR to be installed on the vehicle. Recent bus and communication technologies were evaluated by OEMs to achieve higher rates.

FlexRay is a high-speed bus that uses two communication channels to reduce the risk of failure. Due to its deterministic characteristics, it is suitable for safety applications such as Drive-by-Wire (DBW) systems [17]. Media-Oriented Systems Transport (MOST) is a high-speed media and infotainment module fiber optic network used for high-rate applications. Low-Voltage Differential Signaling (LVDS) is a high-speed differential signaling interface that defines only the electrical layer (receiver and transmitter) and uses twisted-pair copper cables. While it was not developed explicitly for automotive applications, the high bandwidth made possible by LVDS has made it an attractive option for automotive camera manufacturers [18]. It is often supported by automotive-grade devices; however, it will be replaced by Ethernet-based devices in the coming years [19].

To make CAN deterministic, the Time-Triggered CAN (TTCAN) and Flexible Time-Triggered CAN (FTTCAN) were introduced. However, these solutions require that one node in the network perform synchronization, thus violating the multi-primary function of the protocol [20]. In addition, they need a higher-level implementation and hardware. The recent update of CAN (CAN-FD) extends the data range up to 512 bits and the speed up to 2 Mb/s and holds promise for cyber-security and functional applications.

In recent years, Ethernet has emerged as a promising candidate in the automotive industry because it offers high flexibility, high data rate, low latency, ease of installation, and high flexibility. It is also suitable for interconnecting heterogeneous networks via gateways, although strong synchronization is required to compensate for clock drifts in on-board network systems to ensure the Quality of Service (QoS) [21]. Therefore, several sets of Ethernet-based standards have been developed in recent years [22]. For example, Ethernet Audio Video Bridging (AVB) is a real-time networking solution that allows for audio/video traffic to be transported with guaranteed maximum latency [23]. It consists of a set of IEEE 802.1 standards to ensure lag, synchronization performance, and compatibility with legacy Ethernet nodes. Among them, IEEE 802.1Q supports bandwidth reservation requests obtained through the Stream Reservation Protocol (SRP), which allows definition of different classes of traffic. Class A, which provides a maximum latency of 2 ms over seven hops, Class B, offering a maximum latency of 50 ms over seven hops, and best effort traffic, which offers no gurantee [24]. In addition, Time-Sensitive Networking (TSN) technology enhances AVB Ethernet to meet hard real-time requirements with additional Control Data Traffic (CDT).

Similarly, Time-Triggered Ethernet (TTEthernet) is designed for mixed-critical requirements by implementing a time-division multiple-access (TDMA) multiplexing strategy. It distinguishes between three types of flows, Time-Triggered flows (TT), Rate-Constrained flows (RC), and Best Effort (BE), which share the same physical link. Specifically, TT flows are scheduled based on an offline table designed to support global synchronization. Thanks to the pre-designed slot allocation, TT flows can avoid conflicts across the entire network and are suitable for safety-critical applications due to deterministic transmission and predictable delays.

From an architectural point of view, Ethernet-based centralized zonal architectures (Figure 1) are the best solution to combine ADAS and autonomous driving capabilities while maintaining high speed, scalability, simplified wiring harness, low bus load, ISO/SAE 26262 and AUTOSAR compliance, reliability, and power consumption [25,26,27,28]. However, CAN remains commercially mandatory (protection of proven ECU software and hardware, strong electromagnetic immunity (EMC), and reduction in wasted investment already incurred). On the other hand, it is evident that Ethernet-based architectures will become dominant over other high-speed standards due to high BW availability [29], good EMC immunity [30], installation costs, and scalability [31]. A cost-effective solution is to maintain existing CAN (and LIN) vehicles and use Ethernet for new features that require a high data rate [14]. In contrast to the distributed architectures still widely present in the ORVs we know today, in centralized architectures, the logic is moved to Domain Controllers (DCs) which can control a *group* of features.

A further step is the integration of the control logic into a single centralized unit that communicates with hubs located in different areas of the vehicle through a high-speed backbone (e.g., Ethernet) and which in turn communicates with the Input/Output (I/O) slaves, routing the data collected by the vehicle sensors or sending control signals from the central unit to the electro-mechanical actuators (zonal E/E architecture).

#### 1.2.2. Drive-by-Wire Technologies

Traditional mechanical systems, such as those that use a steering column or the direct acquisition of the brake and accelerator by the control units, are not suitable for implementing driver assistance systems and autonomous driving functionalities. The solution is to send a command to the control units in digital format. This way, both a traditional drive by the user and a drive from a control unit are possible.

Drive-by-Wire (DBW) systems replace the mechanical linkage in relation to various vehicle functions, such as the braking, steering and throttle systems, and they are widely used, especially in recent years by vehicle manufacturers. In these systems, vehicle data are collected through sensors and are then processed by control units which translate electrical energy into mechanical movement through appropriate electro-mechanical actuators. There is a number of advantages to using these systems.

For example, Throttle-by-Wire systems (TBW) could improve engine emissions and support features such as torque management with cruise, traction, and stability control to be integrated into the ECM [32].

Brake-by-Wire systems (BBW) allow for designers to reduce the total weight of the entire vehicle and improve energy efficiency, as well as reduce vibration and power loss of the mechanical system. Another advantage of BBW systems is related to scalability, both because the system can be easily adapted to different vehicle platforms and because it is possible to extend it with more sophisticated features such as an Automatic Braking System (ABS), Anti-Split Regulation (ASR), and Vehicle Stability Control (VSC) [33].

Finally, Steer-by-Wire (SBW) systems, characterized by the absence of a steering column, allow better use of the engine, optimize steering response and driver comfort, and eliminate the need for hydraulic fluid in the steering arrangement [34]. DBW technology is fundamental for the implementation of advanced driver assistance features and for the development of autonomous driving, as it allows mechanical actuation employing analog and digital controls.

#### 1.2.3. Architectural Exploration

The identification of both Design Space Exploration (DSE) capability and optimized topologies for Cyber-Physical Systems (CPS) is a crucial point for the future E/E automotive systems development to improve performance, reliability, power consumption, price, and time to market, since unfeasible architectural solutions can be eliminated in the early stages of design [12]. In general, DSE allows for the designer to separate the required functionalities from the architecture and to model them through a graph. To find an optimal (or sub-optimal) architectural solution, optimization techniques are applied considering both feasibility requirements and system requirements. In the literature, there are numerous approaches to DSE for CPS ranging from the aerospace to the automotive sector.

Joshi et al. [35] developed an Integer Linear Programming (ILP) methodology to solve the problem of task schedulability and end-to-end latency requirements satisfiability in an automotive Ethernet network. However, both the multi-domain system feature and the management of different system configuration variants were not considered. A multi-variant-based DSE is proposed by Graf et al. [36] based on ILP for the concurrent synthesis of several ECU variants (HW and SW). In that work, the architectural exploration is fulfilled through the Y-Chart approach, modeling a multicast communication between ECUs.

A further approach for CPS architectural exploration is introduced by Kirov et al. [15] with the ArchEx toolbox. This framework models the architecture through a graph and uses Mixed Integer Linear Programming (MILP) to solve the mapping problem, optimizing the system including timing and reliability constraints. In the same framework, failure probabilities of the system components are used during the design phase to improve the reliability of the overall system [37]. A MILP formulation and solution technique is also used in the wireless sensor network domain to find the optimal topologies considering energy consumption, link quality, and localization requirements, which are all expressed as mixed integer linear constraints over network paths [38,39]. Along these lines, in our prior work, we introduce the idea of using an in-the-loop methodology to reformulate the constraints of Mixed Integer Linear Programming (MILP) problems, but without providing a practical formulation and implementation [40].

Finn et al. [41] employ an iterative optimization-based approach for the design of an aircraft environmental control system. Their experiments demonstrate how to reduce the design time by one order of magnitude when an industrial case study is considered. A DSE for a human intranet network is proposed by Moin et al. [42]. In this work, an iterative approach based on a MILP solver and the discrete-event simulator OMNet++ is used to find candidate solutions satisfying network lifetime and packet delivery ratio.

An architecture modeling and exploration framework for autonomous and semi-autonomous driving is proposed by Zheng et al. [43]. The tool can support architecture evaluation and design, considering the mapping of the software applications onto heterogeneous hardware platforms and different network domains. Several design parameters including timing, reliability, security, and performance are analyzed and optimized.

Drawing upon the existing state-of-the-art and techniques discussed earlier, we apply a novel approach within the off-road automotive domain to address critical challenges related to homologation requirements and the expansion of E/E architectures. In particular, we incorporate cutting-edge technologies and concepts for designing future vehicular networks, including DBW systems, Ethernet standards, and zone-based architectures. Our methodology synergistically combines optimization (for excluding unfeasible architectural solutions) and simulation (for accurate analysis of candidate solutions). To expedite the design process, we employ an in-the-loop approach with a gradient ascent algorithm, iteratively refining constraints until achieving problem convergence to reduce the number of required simulations. This approach significantly enhances the design process, addressing both the inherent time-consuming nature of building computational simulations and the cost and risk associated with constructing physical prototypes.

## 2. Materials and Methods

### 2.1. Methodology

In this subsection, we first recount the ArchEx methodology, emphasizing the basis constraints we take from the original framework. We then describe how to formulate the problem in terms of the parameters of interest and the iterative optimization technique based on in-the-loop simulation (Figure 2).

The system E/E architecture is represented through a directed graph (V,E) in which *V* is a set of components or nodes (such as sensors, actuators, ECUs, etc.) and *E* is a set of edges representing a connection from $v_{i}$ to $v_{j}$ (for example, a CAN bus, Ethernet, etc.), where i,j∈{1,…,|V|} with |V| being the cardinality of *V*. A *re-configurable* vehicular E/E architecture, or *template*, denoted with T, is defined as a graph with a fixed set of components *V* and a variable set of edges *E*. To establish their presence or absence, each edge is associated to a binary decision variable *e*. In particular, $e_{ij}$ is used to denote the connection from component $v_{i}$ to component $v_{j}$. An architectural configuration is an assignment over the variables in *E*. Library L contains all types and attributes related to graph components, which are, in our case, cost, power consumption, weight, dimension, and data rate.

A mapping matrix, denoted as M, associates *virtual* components (graph nodes) to *real* components (physical commercial devices in the library). For each pair of components $v_{i}\in V$ and $l_{j}\in L$, M contains a binary decision variable $m_{ij}$ such that if $v_{i}$ is mapped to $l_{j}$, $m_{ij}$ is equal to one (zero otherwise). When at least one incoming connection $e_{ij}$ or one outgoing connection $e_{ji}$ is equal to one, a component is used (instantiated). Edges can be directly mapped to a set of pre-defined connection elements in the library (i.e., Ethernet, CAN bus, switches) depending on system topology. An ordered sequence of node types $\left(t_{1},\ldots ,t_{n}\right)$ defines a functional flow F (i.e., the link between a sink and a source). Therefore, starting from a template T=(V,E), the optimization is used to find a configuration $E^{*}$ and a map $M^{*}$ such that a set of linear requirements R is satisfied. Given the set of decision variables D=E∪M, the optimal system architecture (topology) is an assignment that includes a subset of components (nodes) and edges (connections) of T and a system implementation (map between nodes to components in L ). The constraints of the system are symbolically represented through patterns to facilitate their specification.

#### 2.1.1. Basis Constraints

**Cost Function.** Each node and edge of T is associated with a cost value (weight, monetary cost, power consumption, dimension, etc.). The sum of all the instantiated component (node) and edge (connection) costs forms a cost function:(1)$$\sum_{i=1}^{\left|V\right|}\delta_{i}c_{i}+\sum_{i=1}^{\left|V\right|}\sum_{j=1}^{\left|V\right|}e_{ij}{\tilde{c}}_{ij}$$
where $c_{i}$ is the cost of node $v_{i}$, ${\tilde{c}}_{ij}$ is the cost of edge $e_{ij}$, and $\delta_{i}$ is a binary decision variable, which is equal to one if the component is instantiated and zero otherwise. Depending on the problem type, some terms in cost Function (Equation 1) can be omitted. To summarize, the weighted sum of different contributions forms the overall cost function, in which particular weights can be set to zero when needed.

**Interconnection Constraints.** Linear arithmetic constraints are used to restrict or enforce valid connections among different nodes. We let *P* be a partition over *V*, such that all components that belong to the same subset in *P* have also the same type. Given *A*, *B* and *C* sets in *P*, the interconnection constraints are expressed as
(2a)$$\sum_{j=1}^{\left|B\right|}e_{a_{i}b_{j}}\geq \left(\leq ,=\right)1\;\forall i\in N:1\leq i\leq \left|A\right|$$
(2b)$$\overset{\left|A\right|}{\underset{i=1}{⋁}}e_{a_{i}b_{j}}\leq \overset{\left|C\right|}{\underset{k=1}{⋁}}e_{b_{j}c_{k}}\;\forall j\in N:1\leq j\leq \left|B\right|$$
where $e_{a_{i}b_{j}}$ is a connection from component $a_{i}$ to component $b_{j}$ (and similarly $e_{b_{j}c_{k}}$). Constraints ([Disp-formula FD2a-sensors-24-04889]) ensure that at least (at most, exactly) one connection from component $a_{i}\in A$ to component in $b_{j}\in B$ exists. Constraints ([Disp-formula FD2b-sensors-24-04889]) ensure that if component $b_{j}$ has a connection to any component in *A*, then it has at least a connection to one node in *C*.

**Mapping Constraints.** We denote with $L_{k}$ and $P_{k}$ the subsets of L and *V* which include all the elements of type *k* in L and *V*. The mapping matrix for type *k* is defined as $m^{k}$, where $m_{ij}^{k}=1$ if a virtual component (corresponding to graph node $v_{j}\in P_{k}$) is implemented by component $l_{i}^{k}\in L_{k}$. Denoting with e the adjacency matrix of T (i.e., $e_{ij}=1$ if a connection from node $v_{i}$ to node $v_{j}$ exists, and zero otherwise), type *k* mapping constraints can be expressed as
(3a)$$\overset{|L_{k}|}{\underset{i=1}{⋁}}m_{ij}^{k}=\overset{\left|V\right|}{\underset{i=1}{⋁}}\left(e_{ij}\vee e_{ji}\right)\;\forall j\in N:1\leq j\leq \left|P_{k}\right|$$
(3b)$$\sum_{i=1}^{|L_{k}|}m_{ij}^{k}\leq 1\;\forall j\in N:1\leq j\leq \left|P_{k}\right|$$

Constraints ([Disp-formula FD3a-sensors-24-04889]) ensure that a mapping between real and virtual components in L exists for all components of type *k* which are instantiated, while Constraints ([Disp-formula FD3b-sensors-24-04889]) ensure that the virtual components are never mapped to more than one library component. This encoding approach helps the exploration of different types of implementations, since a change in L results in only a change in mapping constraints.

#### 2.1.2. Application Constraints

In this subsection, we introduce the new constraints and methodologies used both for the design phase and to analyze the results.

**Rate, Cost, Weight, and Dimension Constraints.** In this work, we are interested in complying with project constraints which are the cost of the system, its weight, the size, and the communication rate of the backbone. We define these system constraints as
$$\sum_{i=1}^{\left|V\right|}\delta_{i}c_{i,t}\leq \lambda_{t},\;t\in \left\{\mathrm{Rate},\mathrm{Cost},\mathrm{Weight},\mathrm{Dimension}\right\}$$
where $c_{i,t}$ is the specific cost value *t* of component $v_{i}$, and $\lambda_{t}$ is the maximum value that the specific cost value *t* is for the entire system (i.e., maximum cost, weight, dimension, etc.).

**CAN Bus Load.** The CAN bus load is calculated through the following equation:(4)$$\mathrm{BL}=\frac{1}{r}\sum_{n=1}^{N}\frac{L_{n}}{T_{n}}=\frac{1}{r}\sum_{n=1}^{N}F_{n}L_{n}$$
where *r* represents the bus baud rate, *N* is the number of CAN bus messages scheduled on the CAN bus, $T_{n}$ is the transmission period of a message, $L_{n}$ is the frame length and $F_{n}$ is the number of frames per second. In other words, $L_{n}/T_{n}$ (or equivalently $L_{n}F_{n}$) is the message bandwidth.

**Camera Data Rate.** In most commercial camera manuals, the equivalent data rate is not reported. To correctly insert the equivalent data rate *r* of each camera in our component library L, we calculate it as
(5)$$r=\frac{H_{px}V_{px}b_{depth}F_{ps}}{\mathrm{CR}}$$
where $H_{px}$ and $V_{px}$ are the number of horizontal and vertical pixels (resolution), $b_{depth}$ is the bit depth, $F_{ps}$ represents the number of frame per second, and CR is the H264 compression ratio.

**Latency and Jitter.** The latency of each stream is measured during simulation, while the jitter (difference between successive latencies) is derived from Equation (Equation 6):(6)$$J_{k}=\left|t_{{sk}_{i}}-t_{{sk}_{i-1}}\right|,i>0$$

**Data Payload.** The required data rate for each stream in the network for the AVB (audio–video bridging) protocol is defined in Equation (Equation 7) [44,45]:(7)$$r=\frac{8\mathrm{MIF}(\mathrm{MFS}+\mathrm{PFO})}{\mathrm{CMI}}$$
where CMI represents the Class Minimum Interval (125 µs for Class A and 250 µs for class B), MIF is the maximum interval frame period (number of frames sent in one CMI), MFS is the Maximum Frame Size, and PFO is the Payload Frame Overhead (Header, Frame Check Sequence, etc.). To perform the simulation, we need to specify the MFS as a parameter. Therefore, after having calculated the equivalent rate from Equation (Equation 5), we retrieve the MFS from Equation (Equation 7).

#### 2.1.3. Gradient Ascent Algorithm

After the optimization phase, simulation is used to check whether the solution is feasible. We use a steepest ascent-based method to reformulate the constraints and feed them back to the optimizer to converge quickly to the optimal solution (Algorithm 1). The two main computational advantages of the gradient ascent algorithm are the simple implementation and the low storage requirements necessary.
**Algorithm 1** Gradient Ascent Algorithm$\alpha_{n}\leftarrow \alpha_{0}$**while**$g_{n}\leq \epsilon$**and**iter<N**do**    Evaluate cost function $x_{n}\leftarrow f\left(x_{n}\right)$    Calculate gradient $g_{n}\leftarrow \nabla f\left(x_{n}\right)=x_{n}-x_{n-1}$    **if** $x_{n}\neq 0$
**then**                                                                                                                ▹Solution Feasible        $x_{n-1}\leftarrow x_{n}$    **else**                                                                                                                           ▹Solution Unfeasible        $\alpha_{n}\leftarrow \frac{\alpha_{n}}{K}$        $x_{n}\leftarrow x_{n-1}$    **end if**    Update $x_{n}\leftarrow x_{n}+\alpha_{n}g_{n}$**end while**

Let us assume we need to find the maximum of function $f\left(x\right),x\in R^{n}$, and $f:R^{n}\to R$. We denote the gradient of *f* by $g_{n}=g\left(x_{n}\right)=\nabla f\left(x_{n}\right)$. We compute a step (represented by *n*) along to a given direction $d_{n}$:(8)$$x_{n+1}=x_{n}+\alpha_{n}d_{n},\;n=0,1,\ldots ,$$
for the gradient ascent method, the search direction is given by
(9)$$d_{n}=g_{n}=\nabla f\left(x_{n}\right)$$
The step length parameter $\alpha_{n}$, also known as learning rate, is adapted at each iteration as follows:(10)$$\alpha_{n}=\left\{\begin{array}{c} \alpha_{n-1}\;\nabla f\left(x_{n}\right)>\epsilon \\ \frac{\alpha_{n-1}}{K}\;\nabla f\left(x_{n}\right)<-\epsilon \end{array}\right.$$
where *K* is a tuning parameter for the algorithm and ϵ is the convergence tolerance. Therefore, convergence is reached when
(11)$$|\nabla f(x_{n})|\leq \epsilon .$$

### 2.2. Tool Structure

Before simulating a candidate network, it is essential to use a toolchain to perform the exploration of the architecture, adopting a generalized process that explores different E/E topologies independently of the simulation and provides maximum flexibility during the design phase.

To achieve this, we developed a CPS exploration optimizer based on the ArchEx tool [37,39], extending and adapting it for the requirements of the project. A MILP-based technique is selected to perform the DSE of the ORV E/E architecture. ArchEx is a MATLAB toolbox that uses YALMIP (https://yalmip.github.io/, accessed on 20 July 2024) as the interface to the solver. However, YALMIP is no longer supported by recent versions of MATLAB, and therefore we rewrote the framework in Python, taking the basis constraints needed for our project and adding the required new ones. Instead of YALMIP, we decided to use the CVXPY Python library to perform MILP modeling and solving (via the CPLEX optimizer). We adapted and extended the framework to deal with a zone-based E/E topology for ORVs, a Drive-by-Wire guidance system, and a heterogeneous network including CAN, AVB, and TTEthernet protocols. The structure of the optimization tool can be summarized as a set of templates that are used to model design requirements.

#### 2.2.1. Optimization Toolbox

The structure of the software, shown in Figure 3, is designed to be flexible and application independent, so that through the use of generic classes it is possible to handle various mapping problems. The class *Problem* provides several routines to formulate and solve architectural exploration problems, such as parsing input files, specifying the component library, creating decision variables, formulating constraints and cost functions, and visualizing result data. The modeling of a generic component through several attributes (type and cost) is implemented in the *Component* class, while the class *Library* is a set of *Component* objects. This class provides methods for library creation and mapping constraints definition given an input specification file. The topology selection and mapping decision variables are stored in general data structures, *AdjacencyMatrix* and *LibraryMapping*. The toolbox input consists of two text files (library and problem description), in which the user specifies general information (types, functional flows, tags), the structure of the model (for example, connection rules), and the requirements of the problem through the patterns (i.e., power consumption, cost, weight, dimension, and rate constraints). The library is arranged to work with several component types, in which each record represents a distinct real component, including both a name and a certain number of attributes. Through tagging, components of the same type can be grouped for domain-specific problems. In our project, we consider a zone-based E/E automotive topology; thus, we distinguish between front-center, center-right, center-left, and rear-center sensors, as well as Ethernet switches based on the vehicle location, placing restrictions on feasible connections. Both types and tags can be used as parameters in the template formulation. The class *Algorithm* interfaces the framework to the solver, providing methods for MILP problem solving. In particular, we use a monolithic optimization approach to obtain the different architectural solutions since we iterate the solution only after reformulating the constraints after the simulation. The patterns are implemented in the *constraintHandler* class, and a zonal architecture-based template is used to implement the extension for ADAS requirements. The inputs of the solver are the cost function and the list of constraints. The results are the computed adjacency matrix, which contains the information about system connections, and the mapping matrix, which contains information about the system implementation.

#### 2.2.2. Simulation Toolbox

From an industrial point of view, simulation is fundamental to evaluate whether the integration of different sensors in the architecture is compliant with the delay requirements or to evaluate the central ECU maximum processing delay. Different domains are generally interconnected through gateways acting as interfaces between different bus technologies, as the coexistence of different protocols over the same vehicle architecture is crucial. First of all, simulation is very important to evaluate those dynamic aspects that the optimizer is not able to model. In addition, we use simulation to validate candidate architectures selected by the optimizer to (i) obtain more accurate results and (ii) guide the optimization (problem constraint re-formulation) along with alternative solutions whenever the requirements are not satisfied. To evaluate mixed-critical network architectures, we select an OMNet++-based framework that supports real-time Ethernet (AVB and TTEthernet), CAN, and FlexRay networks as the most suitable environment to perform simulations [14].

The simulation framework is structured as follows (Figure 3): FiCo4OMNeT implements fieldbus communication, including CAN and FlexRay, while CoRE4INET is an extension to the INET Framework for the event-based simulation of real-time Ethernet in the OMNet++ simulation system, like TTEthernet (AS6802) and IEEE 802.1 Audio Video Bridging (AVB). Finally, the SignalsAndGateways framework uses CoRE4INET, INET 3.6.6, FiCo4OMNeT, and OMNet++ 5.5.1 to enable a heterogeneous network simulation in the OMNet++ environment, including gateway components to enable communication between Ethernet and bus technologies [46]. This environment allows us evaluation of different network metrics such as delay, CAN aggregation effects, and bus load, which are the relevant features for future in-vehicle applications. It also supports the combination of different standards, like the combination of time-triggered traffic and CBS of AVB.

#### 2.2.3. Simulation in-the-Loop (SIL) Manager

After each optimization step, all required simulation configuration files are automatically generated by the SIL Manager. In particular, the initialization files (.ini), the network description files that define the hardware parameters and the network topology (.ned), and the CAN-to-Ethernet gateway configuration file (.xml), which contains both the specification of the CAN messages to be routed over the Ethernet domain and the aggregation strategy, are automatically written from the topology selected by the optimizer. After generating the configuration files, the SIL Manager launches the simulation. The results are then automatically extracted through the opp_scavetool method, provided by the OMNet++ engine, which converts files from the native .vec and .sca formats into .csv. At this point, the results are analyzed, verifying the latency for each data stream, the bandwidth occupation, calculating the jitter and verifying the bus load of the CAN line. To manage the interface between the optimizer and simulation environment, batch Application Programming Interfaces (APIs) are developed. This choice allows independence of the software from the simulator, to make it extensible if other simulation environments are used in the future. Specifically, simulations are run within a MinGW command line, with a bash script that runs the simulation in OMNet++ without a Graphical User Interface (GUI) to save resource usage. We implement automatic communication between the software and the “runner.bash” that manages the simulation by writing and clearing flags that signal the beginning or end of the simulation. To implement both the gateway between the CAN network and the Ethernet domain and the CAN network simulation setup, we use the Pyhton “cantools” library, which parses the CAN database (.dbc) containing all the CAN message features (ID, Transmission rate, etc.). Starting from a list of necessary signals to be forwarded to the ECU that implements the control logic, we automatically generate an .xml file describing the implementation of the gateway, which is necessary to simulate the message forwarded from the CAN domain to the Ethernet domain. In addition, the SIL Manager computes the bus load to compare it with the simulation results and to verify that it meets the system requirements. After verifying the results, if they do not meet the requirements leading to an unfeasible solution (e.g., if the schedulers of the control units are not able to allocate the slots for communication), the constraints are reformulated and re-inserted as input to the optimizer in an iterative manner until a satisfactory solution is obtained. For the reformulation of the constraints, we use a search strategy based on the gradient ascent method to avoid a brute-force approach that is unfeasible due to the high simulation times.

## 3. Results and Discussion

We evaluate our framework by performing a DSE for a zone-based off-road automotive E/E architecture. In particular, the process first involves the selection of hardware components necessary for the implementation of assisted and autonomous driving functions (selecting from a set of RaDARs and/or cameras). The result of the optimizer is both the choice of “virtual components”, i.e., the components instantiated at the architectural level (the decision variables set to one in the adjacency matrix), and the choice of the “real components”, i.e., the commercially available hardware of which has been selected for the implementation of the system (the decision variables set to one in the mapping matrix). The set of commercial sensors available in the library consists of RaDAR and automotive cameras with different characteristics (data rate, cost, size, power consumption, and weight), while all PHY layers are 100BASE-T1 (speed of 100 Mbps within 15 m). Starting from an oversized template and a library of components, the optimizer takes system requirements (bandwidth, cost, size, weight, and power consumption) and an objective function as input, and generates a candidate solution. After the optimization process, the simulation and extraction of the results is performed automatically in order to verify that the system requirements related to latency and load on both the Ethernet backbone and the CAN network are met. If the solution is not feasible (e.g., if the latencies of the various streams are too high, the load on the networks is too excessive, or it is not possible to schedule all the streams at the initialization of the various nodes), the constraints for the optimizer (e.g., the maximum percentage of bandwidth that can be allocated) are reformulated and the process repeats iteratively until it converges to a feasible solution or one that satisfies a specific goal.

### 3.1. Architectural Template and CAN Gateway Implementation

In the problem, the existing CAN architecture is fixed while the high-speed architecture (Ethernet) is selected by the optimizer. In our design, we use Class A AVB streams (guaranteed maximum latency of 2 ms) for the data transmission of sensors at high data rates that perform assisted driving tasks and TTEthernet (deterministic constant latency) to control two DBW modules. The template consists of a central ECU in charge of executing the logic related to the assisted and/or autonomous driving functions (obstacle and pedestrian detection, traffic sign recognition, driver monitoring, and speed assist), and a set of network switches that forward the data from the sensors located in various areas of the vehicle to the central ECU, as well as the control data of the central ECU to the SBW modules that provides steering control. The switches also forward to the central ECU the data converted from CAN to Ethernet which contains various vehicle data (vehicle speed, engine rpm, steering angle, current gear, power-take-off state, transfer case status, etc.) and which are necessary for the implementation of the various ADAS functionalities. Therefore, a significant and mandatory step in this work is the design and definition of the gateway implementation specification. Starting from the homologation requirements (GSR), we derived all the J1939 messages that must be routed from the CAN domain to the Ethernet domain (Table 1). This step is critical for the design of the gateway and for understanding through simulation the impact of CAN routing on the load of the Ethernet network and the latencies of the various streams. These dynamic cross-traffic behaviors (including CBS and message scheduling for different standards) in the vehicular network are difficult to model within the optimizer, so we rely on simulation for their analysis.

### 3.2. System Requirements and Evaluation Setup

Regarding system requirements, the total load of the CAN bus, including new ADAS messages that must be captured from the instrument cluster to alert the driver, must not exceed 50% to ensure that all CAN messages are sent according to their transmission period (to minimize the risk of collisions within the network) [47]. In addition, the maximum latency of each sensor/control stream must be less than 1 ms to ensure that perception tasks are completed up to the required time limit. We therefore set it as the maximum latency limit for streams traveling over Ethernet.

All evaluations are run on a ZBOOK with 32 GB of RAM and an Intel core-i7-11850h. Our first design process executes each simulation for 5 s of simulated time, because it is a number much greater than 0.5 s which is enough to see whether scheduling through the SRP protocol of all streams is feasible.

### 3.3. System Bandwidth Optimization

In the first evaluation of the tool, our goal is to determine the maximum bandwidth constraint for the backbone in order to obtain a feasible solution (all streams and messages can be scheduled and transmitted). As a first hypothesis, we set 75% as the maximum bandwidth limit for non-cross traffic specified by AVB standards. However, the simulator shows that the solution is unfeasible, since the dynamic aspects of the network and electronic integration, such as programming and forwarding CAN messages to the Ethernet, are not modeled by the optimizer. We must therefore lower the maximum bandwidth constraint. To find the maximum bandwidth constraint for which a feasible solution exists, several approaches can be adopted. A brute-force approach is not affordable because of simulation time (5 s of “virtual” simulation requires about 2800 s of “real” time simulation). One alternative is to use linear search. The drawback of this method is a potentially slow convergence rate. A better solution consists instead in modeling the constraint as a cost function and using the gradient ascent-based algorithm (Algorithm 1) to change the bandwidth constraint and converge close to the maximum within a few iterations. This algorithm must be used with care. If the learning rate is too small, convergence is slow. If it is too high, and the function has narrow peaks, the algorithm might hop around the best solution, behaving almost like a random search [48]. In our case, this approach works well, since the bandwidth constraint is an increasing function, which reaches a maximum and then drops to zero when it leads to an unfeasible solution.

Our design solution is found in nine iterations and 25,552 s (approximately 7 h), where most of the time is due to the simulation. This underscores the importance of combining the optimizer and the simulator. The optimizer selects only the most promising architectures, guiding a potentially expensive performance evaluation to much fewer cases. Conversely, the simulation leads the optimizer to exclude solutions that turn out to be infeasible, reducing the search space. The effect is, therefore, a conceivably much faster convergence toward a solution that satisfies the application requirements. The cost function value over the different iterations is shown in Figure 4.

The resulting first architecture and the implementation are shown in Figure 5a, which shows the selected topology, and Figure 6a, providing the correspondence between virtual and real library components. The results show that six RaDARs are selected. Extended simulation data (we launched a more time-consuming simulation only after we found the optimal solution) about latency and jitter are reported in Table 2. From the latency plots (Figure 7a), it is possible to see the effect of the online Credit Base Shaping (CBS) mechanism used in AVB for online bandwidth allocation. The Credit parameter for each stream class is decreased during transmission (send-Slope) and increased otherwise (idle-slope). When credit associated to each SR queue is non-negative, pending frames can be transmitted. The latency of TTEthernet deterministic traffic is instead fixed as expected. In our implementation, the central ECU sends two time-triggered control frames for the SBW modules every 2 ms, one controlling the right wheel and the other the left wheel, each using a bandwidth of 50 kbps (100 kbps total). The deterministic latency is guaranteed by pre-defined slots for the transmission of TT Frames. The introduction of additional messages to satisfy ADAS requirements (information to warn the driver under certain conditions) results in a CAN bus load increase from 32.03% to 48.02%, but within the 50% required bound, and the measured bandwidth occupation in the Ethernet backbone is 63.43%. Finally, the obtained system cost is 6150 EU.

### 3.4. System Power Consumption Optimization

As a second evaluation of the tool, we aim to minimize the power consumption of the entire system. In this case, a feasible solution is found in just one iteration (approximately 45 min), highlighting how solving the previous problem related to bandwidth constraints facilitates the whole process when other project objectives are pursued. The obtained system cost is 5750 EU, while the system electrical power consumption is 17.3 W. The design results are reported in Figure 5b and Figure 6b and Table 3. In this case, the optimizer selects two cameras and two RaDARs. This outcome is consistent with our expectation, because in general, passive components such as cameras consume less electrical power than active components (e.g., RaDAR, LiDAR) since they do not need to emit a signal to probe the environment. It can be observed in Figure 7b that the latency of the streams transmitted by the cameras is higher than that of those transmitted by the RaDARs because larger amounts of data are sent (higher transmission rate), while the latency of the streams transmitted from the central ECU to the SBW modules has not changed. The load on the CAN bus has not been altered because the same architecture and message transmission matrix has been maintained, while the bandwidth occupation on the backbone (connection between the switch and the central ECU, where all data is conveyed) has slightly decreased, to 62.28%.

### 3.5. Effects of CAN Aggregation Strategy

As a third assessment, we use the tool to study the effects of CAN message aggregation (pooling strategy with Hold-Up time). Since the minimum Ethernet payload is 46 bytes and the standard CAN message payload is 8 bytes, direct forwarding of the CAN data results in Ethernet payload padding that wastes bandwidth [14]. Given Hold-Up time, i.e., the maximum acceptable delay for a CAN message, the gateway performs CAN aggregation, releasing frames in groups to the Ethernet domain. More messages are therefore assigned to a shared pool and transmitted trough Ethernet when the Hold-Up time expires. The drawback of using an aggregation strategy is an increase in latency and jitter. The experimental effects of CAN aggregation are analyzed for the first scenario, in which more bandwidth is used. Hold-Up time is set from 0 (CAN message immediately forwarded) to 10 times its transmission period. In the scenario considered, the reduction in channel utilization on the backbone when increasing the Hold-Up factor is completely negligible (the channel utilization varies from 63.43% to 63.26%) because the amount of CAN data to be forwarded is small enough that it does not adversely affect the load on the Ethernet backbone. In addition to the delay in CAN messages, the increase in the Hold-Up factor also increases the latency and jitter of the AVB SR-A classes carrying RaDAR and camera data, as shown in Figure 8. For this reason, the best design choice for the gateway is to immediately send CAN messages over the Ethernet domain without the use of aggregation.

### 3.6. Pareto Fronts

In addition, we evaluated the trade-offs between the system power consumption and the total cost, as well as between the system power consumption and the total weight. We fulfilled this analysis by placing different constraints of weight and cost and minimizing the total consumption of the system. Once again, we emphasize that solving the first problem related to bandwidth constraints allows for us to perform a Pareto analysis, which would otherwise have been very difficult. The results are shown in Figure 9. This analysis is very important from an industrial point of view, because it enables engineers to guide the design not only from a technical perspective, but also taking into account other commercial and marketing business processes or requirements that depend on the supply chain.

### 3.7. Scalability

The last contribution of this work is a qualitative analysis of the possible scalability of this design methodology in the case of system bandwidth optimization, as there are no characterizations of a similar methodology applied to an off-road scenario in the literature. In particular, we tested the tool for larger scenarios but decreasing the simulation time to 0.5 s (which is the minimum reasonable value to see the feasibility of the solution, otherwise it would be impossible in terms of time to complete the characterization).

In the first analysis, we extended the architectural template to support up to 32 sensors, thus saturating the bandwidth in three different cases by setting a maximum of 100 Mbps, 1 Gbps, and 10 Gbps, respectively (the greater the capacity of the link, the greater the amount of data transmitted, and consequently the longer the time required to complete a simulation). We then retrieved the iterations and the total resolution time obtained with our method and compared them with the hypothetical iterations and the relative time required by a simple linear search (both this approach and a brute force one would be impossible to implement given the large time required for simulations). The iterations required by a linear search are calculated by reasonably assuming that the initial bandwidth constraint is set to a fraction of the maximum link capacity and then incremented to the value found with our tool. The total time for the linear search was calculated by taking the average simulation time obtained with our tool multiplied by the number of iterations. Comparative results for iterations and total resolution time are shown in Figure 10. The time and iterations for solving the problem are shorter with our methodology and significantly smaller than the linear search method as the maximum bandwidth capacity of the system increases.

In the second analysis, we tested the maximum resolution time by setting the maximum number of sensors in the template from a minimum of 4 to a maximum of 32, and having as maximum link capacity of 100 Mbps and of 1 Gbps (Figure 11). The total resolution time grows as the number of sensors increases and when the maximum link capacity is higher. However, it is still limited and acceptable from a computational resources perspective, considering that a number of 30 (or more) sensors are required for projects which must satisfy SAE Level 3 for autonomous driving (https://semiengineering.com/how-many-sensors-for-autonomous-driving/, accessed on 20 July 2024).

## 4. Conclusions

In this work, we combined optimization-based design-space exploration with event-driven simulation, optimizing abstract architectural models using Mixed-Integer Linear Programming (MILP) algorithms and providing detailed insights into system behavior. In addition, we developed a methodology which identifies feasible candidate architectures through optimization, evaluates them using a simulator in an iterative loop, and incorporates extra constraints to meet system requirements during optimization. Our study focused on the automatic design of electrical/electronic (E/E) architectures, specifically for off-road vehicles, a niche area not yet extensively studied. Finally, we demonstrated the validity of the approach through optimal results which are achieved after only a few iterations. An equivalent manual exploration of the design space would take days of trials and errors.

Although the approach was developed for a specific domain of study under our expertise, the methodology is of much larger scope and could be easily extended to the commercial vehicles sector that increasingly faces similar challenges. Indeed, the issues that gravitate around safety and autonomy of off-road vehicles today will be common to the transportation industry at large, requiring a shift in the design methodology to cope with stringent requirements. The method developed in this paper is easy to adapt to different scenarios by adjusting the architectural template, implementing a new forward strategy for the CAN gateway, and adding problem-specific constraints while taking into account the data exchanged within the vehicle. The main strength of the methodology lies in the patterns, which concisely summarize hundreds if not thousands of linear and linearizable constraints in a single line, helping application domain experts develop a formal specification. The challenges lie in the computational complexity, which iterative approaches like the one introduced in this work can help cope with. From an industrial viewpoint, the creation of automatic design tools is highly significant because they exclude in advance the architectural solutions that are not feasible, thus saving money and time that would be required to create physical prototypes. In the future, we will study how to integrate other cyber-security and safety requirements into the tool, as well as testing and comparing the architectural solutions obtained with a real vehicle.

## Figures and Tables

**Figure 1 sensors-24-04889-f001:**
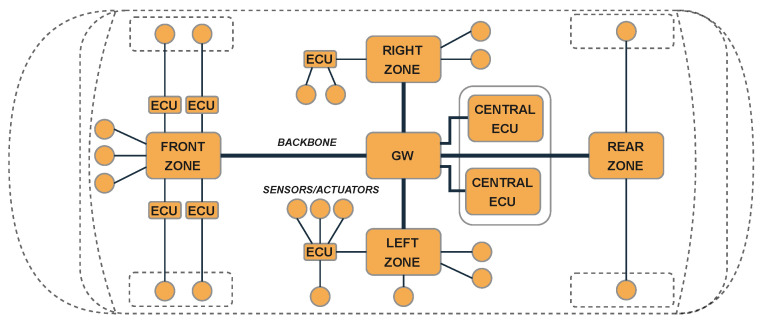
Example of zonal-based E/E vehicle architecture.

**Figure 2 sensors-24-04889-f002:**
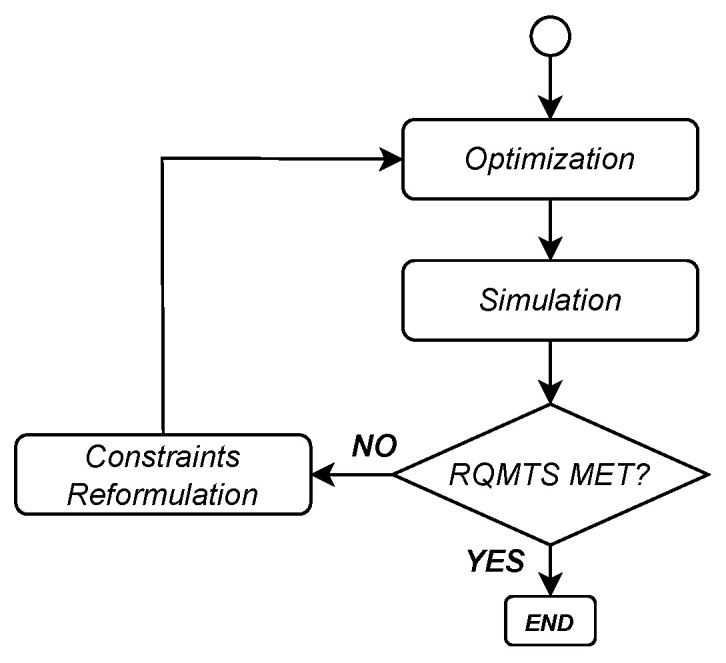
Conceptual chart of the design methodology for the E/E Exploration which combines optimization and simulation.

**Figure 3 sensors-24-04889-f003:**
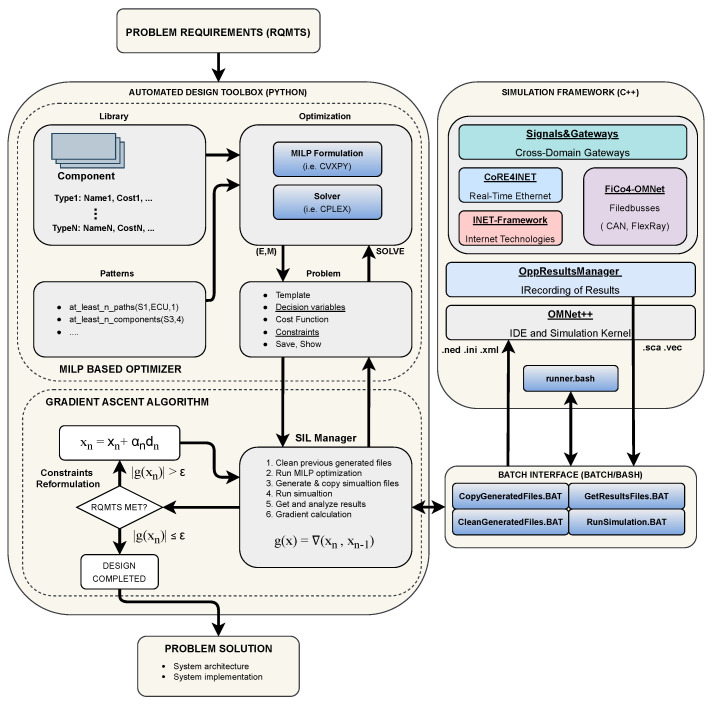
Framework for automated design of ORV mixed-critical in-vehicle networks.

**Figure 4 sensors-24-04889-f004:**
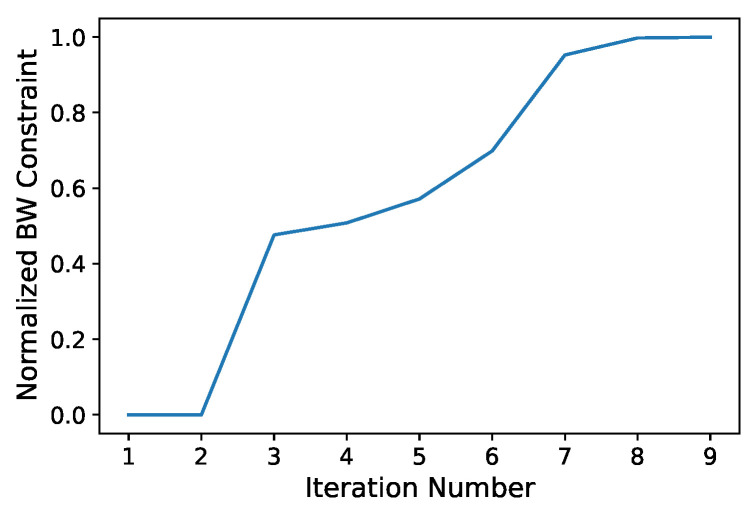
Bandwidth constraint during design iterations using the gradient ascent method.

**Figure 5 sensors-24-04889-f005:**
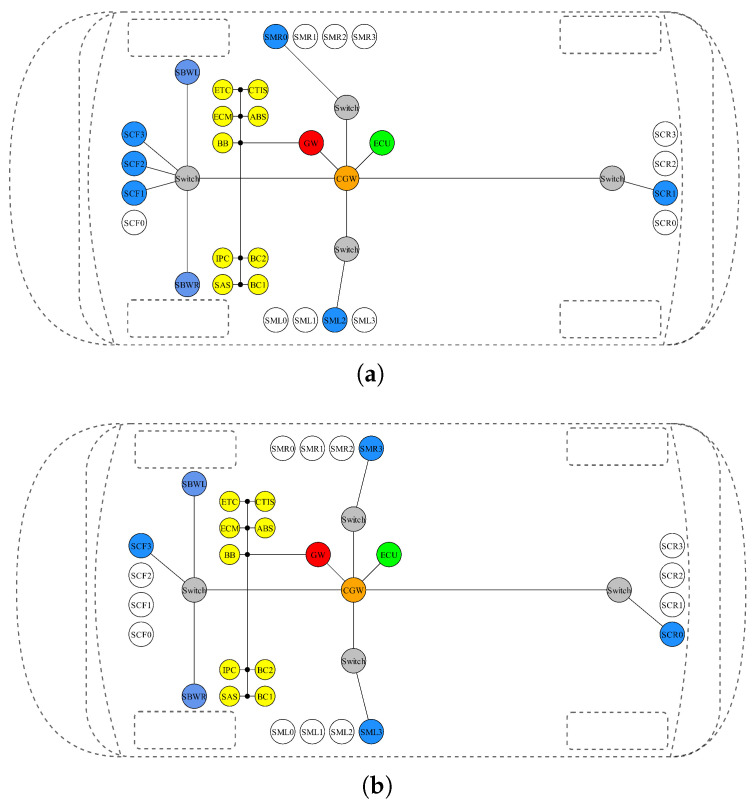
Optimized E/E graphs showing the instantiated virtual components. Colored components are those selected by the optimizer. (**a**) bandwidth maximization, (**b**) energy minimization.

**Figure 6 sensors-24-04889-f006:**
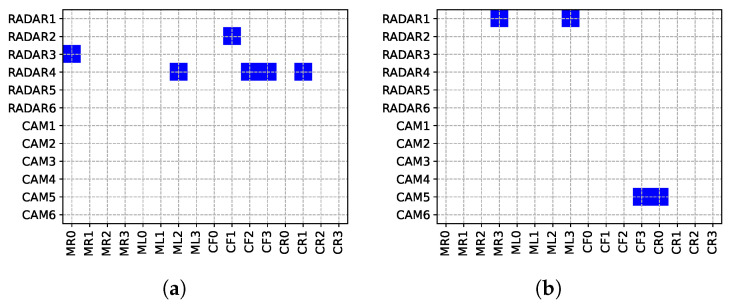
Optimized mapping matrices between virtual and real components representing the commercial devices selected as sensors. (**a**) bandwidth maximization, (**b**) energy minimization.

**Figure 7 sensors-24-04889-f007:**
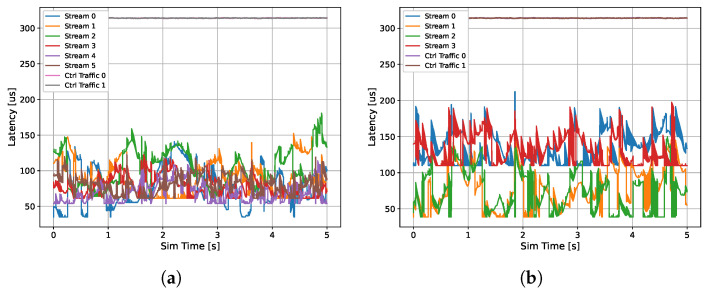
Optimized E/E architecture stream latencies of the different simulated streams (AVB and TTEthernet) of sensor/command data. (**a**) bandwidth maximization, (**b**) energy minimization.

**Figure 8 sensors-24-04889-f008:**
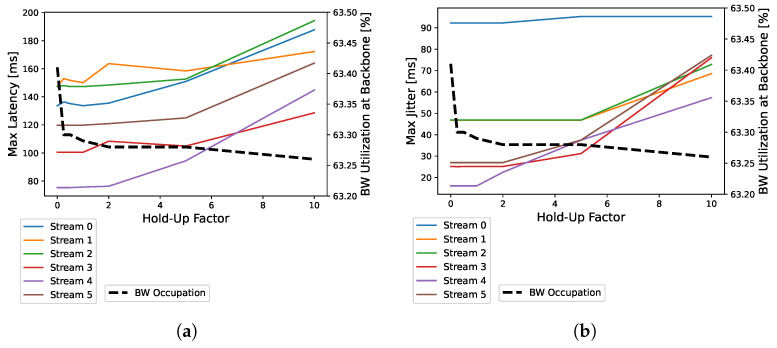
Latency and jitter analysis for different hold-up factor values. (**a**) maximum latency vs. hold-up factors, (**b**) maximum jitter vs hold-up factors.

**Figure 9 sensors-24-04889-f009:**
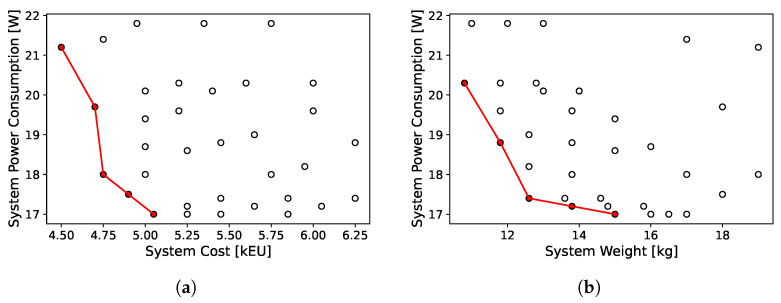
Pareto fronts in the case study for optimizing system power consumption (the red line is the Pareto front). (**a**) system power consumption vs system cost, (**b**) system power consumption vs system weight.

**Figure 10 sensors-24-04889-f010:**
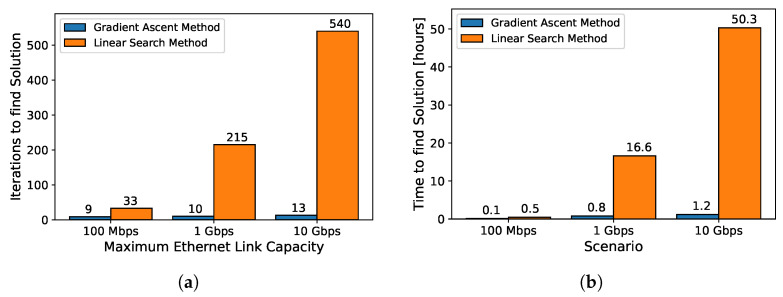
Scalability analysis for different maximum link capacity scenarios between gradient ascent and linear search methods. (**a**) maximum link capacity vs total iterations, (**b**) maximum link capacity vs total resolution time.

**Figure 11 sensors-24-04889-f011:**
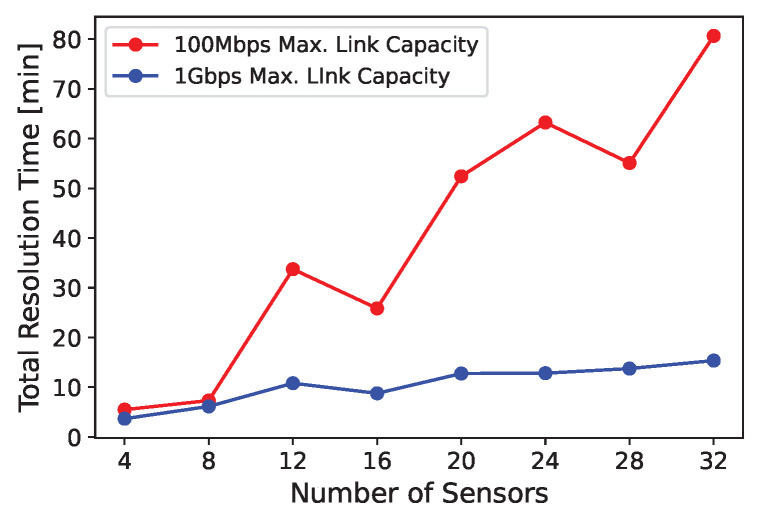
Number of sensors and connection capabilities vs total resolution time.

**Table 1 sensors-24-04889-t001:** Gateway forward table: J1939 CAN messages (information and transmission rates) which need to be routed to the Ethernet domain to meet GSR requirements (SAS and TCO are the steering angle sensor and tacograph ECUs, respectively).

Node/Function	GSR Requirement	J1939 Msg Name	Information	Tx Rate [ms]
ABS	DDAW, REIS	EBC1	Brake Switch Information	20
ABS	AIF, BSIS, DDAW, ISA, MOIS, REIS	EBC2	Vehicle Speed	100
ABS	DDAW	VDC2	Yaw rate	100
ABS	AIF, BSIS, DDAW, ISA, MOIS, REIS	HRW	Vehicle Speed	20
TCO	AIF, BSIS, DDAW, ISA, MOIS, REIS	TCO1	Vehicle Speed	20
TCO	AIF, BSIS, DDAW, MOIS, REIS	VDHR	OdometerData	1000
TCO	AIF, BSIS, DDAW, MOIS, REIS	TD	Time and Date	1000
ECM	AIF, DDAW, MOIS, BSIS, REIS	CCVS1	Clutch Status, Parking Brake status	100
ECM	AIF, DDAW, REIS	EEC2	Throttle Switch Status	100
ECM	AIF, DDAW, ISA, REIS	EEC1	Engine Speed	10
ETC	DDAW, BSIS, MOIS, REIS	ETC5	Rear gear Information	1000
ETC	DDAW, REIS	PTO	Power Take-Off information	100
ETC	DDAW, REIS	ETC1	Power Take-Off information	10
ETC	DDAW, BSIS, ISA, MOIS	ETC2	Gear Information	100
BCM	BSIS, DDAW, MOIS, REIS	LD	Lighting Data	50
BCM	DDAW	EAC1	Differential Locks State	500
BCM	BSIS, DDAW, MOIS, REIS	TCI	Transfer Case State	1000
SAS	DDAW	SAS	Steering Wheel Angle	10

**Table 2 sensors-24-04889-t002:** Extended Simulation results of the optimized E/E architecture for maximized BW constraint.

Traffic	Src	Dst	BW [Mbps]	Protocol	Max Latency [µs]	Mean Latency [µs]	Max Jitter [µs]	Mean Jitter [µs]
Stream 0	CF1	ECU1	5	AVB Class-A	140.75	76.77	92.31	0.41
Stream 1	CF2	ECU1	12	AVB Class-A	152.25	89.23	62.83	0.51
Stream 2	CF3	ECU1	12	AVB Class-A	180.87	100.79	51.91	0.46
Stream 3	ML2	ECU1	12	AVB Class-A	122.53	77.09	46.25	0.48
Stream 4	MR0	ECU1	10	AVB Class-A	113.96	68.16	46.20	0.48
Stream 5	CR1	ECU1	12	AVB Class-A	123.07	78.97	47.67	0.48
Ctrl Traffic 0	ECU1	SBWR	0.05	TTEthernet	315.11	314.06	0.93	0.24
Ctrl Traffic 1	ECU1	SBWL	0.05	TTEthernet	314.93	314.06	1.03	0.25

**Table 3 sensors-24-04889-t003:** Extended Simulation results of the optimized E/E architecture for minimized energy consumption.

Traffic	Src	Dst	BW [Mbps]	Protocol	Max Latency [µs]	Mean Latency [µs]	Max Jitter [µs]	Mean Jitter [µs]
Stream 0	CF3	ECU1	25	AVB Class-A	212.08	130.45	101.97	1.25
Stream 1	ML3	ECU1	6	AVB Class-A	148.55	79.97	98.13	0.89
Stream 2	MR3	ECU1	6	AVB Class-A	149.85	74.43	98.08	1.00
Stream 3	CR0	ECU1	25	AVB Class-A	197.48	128.70	87.49	1.26
Ctrl Traffic 0	ECU1	SBWR	0.05	TTEthernet	315.06	314.07	1.05	0.25
Ctrl Traffic 1	ECU1	SBWL	0.05	TTEthernet	314.91	314.03	0.91	0.25

## Data Availability

The data presented in this study are available on request from the corresponding author due to confidentiality reasons.

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
