# Peer review of "Combining Optimization and Simulation for Next-Generation Off-Road Vehicle E/E Architectural Design"

_sensors, 2024, doi:10.3390/s24154889_

Round 1

Reviewer 1 Report

Comments and Suggestions for Authors

1. It is recommended that the purpose of the study and the reason for the combination of optimization and simulation be summarized more clearly at the end of the introduction.
2. Constraints and algorithm of 2.1.2 are mixed and described, making it easier for readers to understand, so the algorithm part needs a separate description. In addition, please use the paper standard form number instead of the bold font.
3. In Figure 3, it is recommended that the optimization toolbox and the simulation toolbox be expressed or modified as a boundary line because there is only CPLEX solver in the algorithm part and it is difficult to know the flow in which bandwidth or energy consumption is optimized.
4. It was easier for readers to understand if the conclusion was summarized more clearly from a contribution point of view.
5. Terms that have not been used repeatedly are also written in abbreviations and need to be checked.

Reviewer 2 Report

Comments and Suggestions for Authors

The submitted manuscript titled "Combining Optimization and Simulation for Next-Generation Off-Road Vehicle E/E Architectural Design" introduces an automated approach for designing E/E architectures in off-road vehicles. The authors employed a MILP-based optimizer to identify potential solutions, validated these using a discrete event simulator, and utilized an ascent-based gradient method to refine constraints, ensuring convergence to the final architectural solution. The manuscript is well written; however, I have a few minor concerns:

  1. Line 182: Please consider providing the full forms of acronyms such as ABS, ASR, and VSC for clarity.

  2. It is advisable to ensure that each table and figure (e.g., #3, #4, #8, and #10) is appropriately introduced and explained with its corresponding number in the text before insertion into the manuscript. This approach will enhance readers' comprehension of the figures.

  3. The conclusion could benefit from further exploration of extending the methodology to the commercial vehicles sector. Delving into specific benefits, challenges, and adaptations needed for different industries would enhance its relevance and applicability.
